# User-Guided Design of a Digital Tool for Health Promotion and Radiation Protection: Results from an Internet Needs Survey

**DOI:** 10.3390/ijerph182212007

**Published:** 2021-11-16

**Authors:** Yui Yumiya, Takashi Ohba, Michio Murakami, Hironori Nakano, Kenneth E. Nollet, Aya Goto

**Affiliations:** 1Center for Integrated Science and Humanities, Fukushima Medical University, Fukushima 960-1295, Japan; agoto@fmu.ac.jp; 2Department of Public Health and Health Policy, Graduate School of Biomedical and Health Sciences, Hiroshima University, Hiroshima 734-0037, Japan; 3Department of Radiation Health Management, Fukushima Medical University, Fukushima 960-1295, Japan; tohba@fmu.ac.jp; 4Department of Health Risk Communication, Fukushima Medical University, Fukushima 960-1295, Japan; michio@fmu.ac.jp; 5Department of Epidemiology, Fukushima Medical University, Fukushima 960-1295, Japan; h-nakano@fmu.ac.jp; 6Department of Blood Transfusion and Transplantation Immunology, Fukushima Medical University, Fukushima 960-1295, Japan; nollet@fmu.ac.jp

**Keywords:** eHealth literacy, risk communication, Fukushima nuclear accident, radiation protection, health promotion

## Abstract

Background: Digital tools can be powerful and effective in connecting people with life-saving and health-promoting support when facing a health crisis. To develop a digital application for radiation protection and health promotion for evacuees returning home after the Fukushima nuclear accident, we conducted a needs assessment survey and explored the association of people’s eHealth literacy (eHL) level with their digital tool knowledge, attitudes, and practice (KAP). Methods: From 339 responses to an online survey, data from 264 lay persons were analyzed. The KAP items were those used in a prior EU project, and eHL levels were assessed with a Japanese version of the eHealth Literacy Scale. Results: Multivariable analyses showed significant associations between eHL and the digital tool KAP for radiation protection (knowledge: adjusted odds ratio (aOR) = 1.10; attitude: 1.06; practice: 1.10) and for health promotion (knowledge: aOR = 1.13; attitude: 1.06; practice: 1.16). Conclusions: People with a higher eHL had a more positive KAP. For those with a lower eHL, we are formulating easy-to-understand explanations to promote the utilization of the digital tool and enthusiasm for future community-oriented digital tools.

## 1. Introduction

### 1.1. Background

Digital tools are becoming increasingly integrated into our daily lives. Especially in the context of disasters, they may be essential to convey emergency information and to connect people with support services. Moreover, eHealth could be helpful in raising public awareness and in providing health services or health education, even when people and providers are geographically separated. A previous scoping review showed that digital eHealth could improve the response to healthcare demands before, during, and after disasters [1]. Augusterfer and colleagues described cases of using digital health in post-disaster settings [2]. With an increase of global disasters, digital health services, including telemedicine, could improve post-disaster mental healthcare for under-served populations [2]. Advances in digital technology, including mobile phones and tablets, enable global real-time connections, allowing medical care to be delivered swiftly across geographic boundaries. Furthermore, digital and mobile health could facilitate the continuous clinical monitoring of people with chronic conditions in routine medical practice. If such services are widely used in the aftermath of a disaster, continuity of care can proceed with minimal disruption to pre-disaster care plans [1]. According to the Health Emergency Disaster Risk Management (Health EDRM) Framework built by the World Health Organization (WHO) in 2019, assessing, communicating, and minimizing risks across the continuum of prevention, preparedness, readiness, response, and recovery, as well as increasing the resilience of communities, countries, and health systems, are all important components [3]. Disaster eHealth approaches should thus be incorporated more frequently into disaster planning and preparedness so as to be ready for use during and after a disaster.

### 1.2. Project Background

As a part of nuclear disaster preparedness, the EU project “SHAMISEN-SINGS: Nuclear Emergency Situations–Improvement of dosimetric, Medical And Health Surveillance–Stakeholder INvolvement in Generating Science” publicized the following recommendations for developing a digital application: (1) optimize content with stakeholders, (2) balance content, security, and development cost, (3) develop a user support system, (4) apply incentives to promote usage, (5) respond to queries about radiation and health, (6) involve vulnerable populations, (7) accommodate multiple languages, and (8) consider ethical issues, especially privacy protection [4]. Building on the SHAMISEN-SINGS project, we launched a project funded by the Japanese Ministry of the Environment in order to develop an application in Fukushima. Its planning follows the SHAMISEN-SINGS recommendations, with an aim to facilitate interactive communication among evacuees (returning home after the Fukushima nuclear accident) and local health workers [5].

### 1.3. Study Purpose

Our survey’s purpose was to identify application needs of the general public by assessing the relationship between eHealth literacy (eHL, “the ability to seek, find, understand, and appraise health information from electronic sources and apply the knowledge gained to addressing or solving a health problem” [6]) and KAP (knowledge, attitude, and practice) in relation to a digital tool for radiation protection and health promotion. A KAP survey is “a representative study of a specific population to collect information on what is known, believed, and done in relation to a particular topic” [7]. Such surveys are often utilized to develop public health interventions. We began by assessing the distribution of eHL scores in a general public cohort, and then the association between the eHL score and KAP in relation to an application for radiation protection and health promotion. Recent systematic reviews indicate the importance of making eHealth interventions easily accessible to low eHL individuals and improving their eHL level [8,9]. Often, however, a careful needs assessment of information to be distributed is omitted, as was the case in the immediate aftermath of the Fukushima nuclear accident. Therefore, the present survey tried to better accommodate a wider range of information needs of community residents in a post-disaster phase.

## 2. Materials and Methods

### 2.1. Study Participants

This online survey adapted participant allocation and survey items from the SHAMISEN-SINGS project protocol. The survey was conducted from 31 January to 4 February 2020 with participants recruited from monitors registered with an Internet survey company, INTAGE Research Inc. Monitor recruitment based on the stakeholder state (lay persons or potential stakeholders in the case of a nuclear emergency [i.e., public servants, medical professionals, teachers]), residential area (area affected by the Fukushima Daiichi Nuclear Power accident, or area not affected, including within and outside a 30 km radius of another nuclear plant), and age (20–29 years old, 30–59 years old, or 60 years and over) (Table 1). In total, 339 responded, and we extracted the data of 264 lay persons for analyses in the present study, because our research aim was to investigate the application needs of the general public in order to address those needs in a subsequent application development.

INTAGE Research Inc. (Tokyo, Japan) is one of Japan’s largest research firms, engaging over 3 million survey participants each year. This company can collect the data by residential area, occupation, and age range for a target group. To ensure the reliability of the monitors, the company implements survey participants’ identification via mail-out and the elimination of poor-quality data such as excessively short response times and multiple responses from the same IP address. In addition, they incentivize participation with tokens such as Internet reward points.

### 2.2. Survey Items

The SHAMISEN-SINGS survey items were first developed in English, then translated and tested in Japan at a smaller scale before conducting the present Internet survey. In addition to the SHAMISEN-SINGS survey items, we included an eHL measure. We asked the KAP items on radiation protection and health promotion separately in the SHAMISEN-SINGS survey as follows, with alternative wording delimited in brackets. 

Knowledge: “Are you aware of existing mobile apps or personal devices that allow you to [perform your own radiation dose measurements/monitor your health status]?” Attitude: “Would you be interested in using a mobile app that allows you to [measure radiation/monitor your health status and well-being] during and after a radiation accident?” Practice: “Have you ever used any mobile apps or devices for [radiation measurements/monitoring your health status]?” Answer options were “yes” or “no”.

As for the eHL level assessment, we used a Japanese version of the eHealth Literacy Scale as a highly validated and reliable scale, composed of eight items [10]. The following questions were asked: (1) “I know what health resources are available on the Internet,” (2) “I know where to find helpful health resources on the Internet,” (3) “I know how to find helpful health resources on the Internet,” (4) “I know how to use the Internet to answer my health questions,” (5) “I know how to use the health information I find on the Internet to help me,” (6) “I have the skills I need to evaluate the health resources I find on the Internet,” (7) “I can tell high quality from low quality health resources on the Internet,” and (8) “I feel confident in using information from the Internet to make health decisions.” Each item was rated on a 5-point Likert scale ranging from “strongly disagree” (1) to “strongly agree” (5), and the sum yielded an eHL total score ranging from 8 to 40.

Radiation-related items included the residential area (within or beyond a 30-km radius of a nuclear power plant) and concern about living close to a nuclear power plant. The latter asked, “Are you personally concerned about potential dangers and risks related to living close to a nuclear power plant?” Those who answered “yes” and “sometimes” were categorized as having concerns. Socio-demographic items included sex, age, employment status, occupation, residential area (inside or outside Fukushima Prefecture), and family structure. Socio-demographic items were selected based on previous studies showing that the eHL level was associated with sex, age, employment status, marital status, and income [10,11].

### 2.3. Statistical Analysis

Chi-square, Fisher exact, and independent t-tests were performed to investigate differences in individual attributes, radiation-related items, and eHL levels associated with KAP for radiation and health assessment digital tools. In addition, Cronbach’s coefficient alpha was used to measure the internal consistency of the 8 items in relation to the eHL level. We then employed a binary logistic regression analysis to investigate bivariate associations between eHL and KAP by controlling for other items that were significant in univariate analyses. The job type was excluded in the multivariable analysis because of the different numbers of total respondents to the other items. Multicollinearity was assessed by using the variance inflation factor (VIF). The VIF for all explanatory variables was less than 3.0. All analyses were conducted using IBM SPSS Statistics 25. A P value of 0.05 was the threshold for statistical significance.

## 3. Results

### 3.1. Descriptive Findings

Table 2 shows the individual attributes of survey participants and radiation-related items. More than 80% of respondents felt concern related to living near any nuclear power plant (NPP). Table 3 shows the eHL levels. The item on how to use health information to answer health questions through the Internet had the highest mean among all eight items, whereas the lowest mean was observed for the skills of evaluating health resources on the Internet. The internal consistency of the eight items was adequately high (Cronbach’s α = 0.91). 

### 3.2. Analytical Findings

Table 4 shows factors associated with “knowledge, attitude and practice” related to a digital application for radiation protection. Those with knowledge comprised 30.7% of the total study respondents; those with a positive attitude comprised 72.3%, and those with practice experience 11.7%. Furthermore, the result showed that a higher eHL was significantly associated with a positive KAP. Regarding knowledge, significant differences in sex, age, residential area, and living were observed. A positive attitude was significantly associated with concern related to living near any NPP, while practice was associated with the sex, age, and residential area. Table 5, with multivariable analysis results, shows significant associations between eHL and KAP even after controlling for other items that were significant in the univariate analyses (knowledge: adjusted odds ratio (aOR) = 1.10, 95% confidence interval (CI) 1.04–1.16; attitude: aOR = 1.06, 95% CI 1.01–1.12; practice: aOR = 1.10, 95% CI 1.01–1.19). Additionally, the results revealed that those living in Fukushima were likely to have more knowledge compared to those outside Fukushima (aOR = 2.17, 95% CI 1.21–3.91), while those who had concern regarding living near any NPP tended to show a positive attitude (aOR = 2.18, 95% CI 1.11–4.29). More men and those living in Fukushima were likely to use such a digital application for radiation protection (sex: aOR = 3.07, 95% CI 1.23–7.69; residential area: aOR = 6.73, 95% CI 2.85–15.94).

Turning to health promotion, Table 6 shows that those with knowledge, attitude, and practice were 18.9%, 65.5%, and 6.8%, respectively. Like the radiation protection application, the results showed that a higher eHL was significantly associated with a positive KAP. Significant differences in education and job types (knowledge), residential area and two radiation-related items (attitude), and living near any NPP (practice) were observed between the positive and negative KAP. Table 7 shows significant associations between eHL and KAP, controlling for other items (knowledge: aOR = 1.13, 95% CI 1.06–1.20; attitude: aOR = 1.06, 95% CI 1.01–1.11; practice: aOR = 1.16, 95% CI 1.05–1.29). Furthermore, the results showed that those who had concern regarding living near any NPP tended to show a positive attitude (aOR = 3.17, 95% CI 1.59–6.31), and those with or without concern living near any NPP were likely to use such a digital application for health promotion (living near any NPP: aOR = 3.18, 95% CI 1.16–8.71).

## 4. Discussion

Despite high positive attitudes toward a digital application for both radiation protection and health promotion, just 11.7% and 6.8% of respondents actually used the tool for radiation protection and health promotion, respectively. Our main finding is that even among Internet survey registrants, a higher eHL level was associated with a positive KAP related to the application. This result is consistent with a previous study indicating that digital literacy, as measured by the frequency of engaging in six activities (visiting blogs, participating in discussion forums, playing games, downloading or listening to music, downloading software, or emailing with friends), was significantly higher in the high eHealth literacy group than in the low eHealth literacy group [11]. Likewise, another study in Japan reported that 81% of the higher eHL group searched health information via the Internet, whereas it was only 49% in the lower eHL group [10]. The same study also found that the high eHL group used more information resources and received a wider range of eHealth information [10]. As a result, those with a higher eHL had better information search results, a better understanding of their health condition, symptoms and treatment options, self-management and change of health behavior, and interaction with their health care professionals [11]. Those with a lower eHL level, on the other hand, are likely to face hurdles when trying to utilize digital tools for health promotion. Therefore, our application project faces the challenge of considering what content and functions would be useful in order to reduce health risks, promote health, and narrow inequalities in terms of access to digital health information between subpopulations with a high and low eHL. 

The reported characteristics of people with a lower eHL were the elderly (over 65 years old) and those with a lower income, less education, and chronic illnesses, who were in general considered as vulnerable [11,12]. In our study, having a higher eHL level, being male, and having a higher education were associated with a positive KAP related to digital tools. Age was associated in the univariate analysis, but its significance diminished in the multivariable analysis. The previous study reported that hybrid methods, which combined online and offline techniques, were crucial for ensuring that vulnerable populations were included [12].

Respondents who lived in Fukushima had a better knowledge and experience of using a digital tool for radiation measurement than those living outside Fukushima, which we attribute to having experienced the Fukushima nuclear accident in 2011. Since that disaster, each municipality in Fukushima has been loaning out personal dosimeters which measure external radiation exposure, e.g., the D-shuttle^®^ (a cumulative electronic personal dosimeter) [13], free of charge, and “radiation counselors” designated by the Ministry of the Environment have been supporting the usage of these measurement tools and responding to various daily life problems, especially with regard to returning and settling in areas where evacuation orders were lifted [14]. Having such support at hand, residents in Fukushima are more knowledgeable about personal dosimetry as opposed to residents outside Fukushima. Furthermore, regarding attitudes toward digital tools, a concern related to living near any NPP was associated with positive attitudes toward a digital application for both radiation protection and health promotion. Since living near any NPP causes great anxiety about an invisible health risk, using a personal dosimeter that can quantify, visualize, and evaluate the risk could contribute to preventing health hazards from excessive radiation exposure and to reducing radiation anxiety [15]. Similarly, those who live near any NPP are using the health application more than those living elsewhere, presumably because they are more concerned about the health hazards of radiation exposure than those who do not live near any NPP. Regarding differences by sex, the current study found that men were more likely to use such a digital application for radiation protection when compared to women. According to a survey of college students from 29 nations, early adopters of new technology are overwhelmingly men [16]. The gender gap in digital skills emerges across regions and economic levels, but it is especially pronounced among women who are older, less educated, poor, or who live in rural or developing countries [16]. Such gaps warrant special attention as societies aspire to be more inclusive.

Our overall findings suggest that approaches need to be adapted to personal, family, community, and national circumstances, in order to effectively provide digital health information to under-served populations. For example, at the family level, we can encourage family members to engage with one another across generations, to provide hybrid solutions allowing those who are more technologically literate to assist individuals who are less so. At the community level, we can train community health workers to be information providers as well as technology “translators” [12]. In a health system, healthcare professionals should provide guidance to patients so that they are aware of and capable of using telehealth services [12]. For the societal promotion of digital health information, comprehensive and inclusive efforts that engage not only those with a lower eHL, but also those with a high eHL and health professionals themselves, are needed.

This study did have some methodological limitations. First, because this was a cross-sectional study, we could not determine any causality between eHL and KAP in relation to a digital application for radiation protection and health promotion. Second, there is a possibility of a sampling bias due to the data collection by an online survey system. The current study revealed no significant differences between age and KAP, partly because online survey registrants, even those who were 60 or more years old, were likely to be skillful in using the Internet and have a higher eHL. Therefore, we are currently conducting face-to-face field interviews using a cultural anthropologist to gather thorough feedback from users of our new application. At the same time, this could also strengthen our study findings insofar as even among those with some familiarity with the Internet, eHL plays a significant role in defining one’s KAP with regard to digital tools.

## 5. Conclusions

In summary, our findings indicate that people with a higher eHL are likely to have a better KAP in relation to a digital application for radiation protection and health promotion. In order to have highly positive attitudes toward a digital application result in its more frequent usage, concerted efforts are needed in order to provide a system that includes community residents at different eHL levels and health professionals, so that everyone can readily access a newly introduced digital tool for radiation protection and health promotion.

## Figures and Tables

**Table 1 ijerph-18-12007-t001:** The Internet survey’s participant allocation and grouping.

Group Category	Occupation	Residential Area	Age Range	Expected Number of Subjects
Lay persons	-	Residents affected by FDNPP ^1^ accident (Hamadoori ^2^ and Nakadoori ^3^ in Fukushima)	20–29 years old	25
	-		30–59 years old	25
	-		60 years and over	25
	-	Residents living within 30 km of another NPP ^4^ in Japan	20–29 years old	25
	-		30–59 years old	25
	-		60 years and over	25
	-	Residents living 30 km or further away from another NPP ^4^ in Japan	20–29 years old	25
	-		30–59 years old	25
	-		60 years and over	25
Stakeholders	Public sevants		-	20
	Medical professionals		-	20
	Teachers		-	20

^1^ Fukushima Daiichi Nuclear Power Plant; ^2^ Where FDNPP is located; ^3^ An area that received nuclear fallout from FDNPP, ^4^ Nuclear Power Plant.

**Table 2 ijerph-18-12007-t002:** Characteristics of research participants.

	N (%) 264 (77.9)
**Individual attributes** Sex	
Female	133 (50.4)
Male	131 (49.6)
Age ^1^	
20–59	178 (67.4)
60 or older	86 (32.6)
Education	
High school or lower	100 (37.9)
Junior college, technical school	64 (24.2)
University, graduate school	100 (37.9)
Employment status	
Employed	172 (65.2)
Unemployed	92 (34.8)
Job type ^2^	
Engineering, technical expert, managerial work	48 (28.7)
Administration	42 (25.1)
Other	77 (46.1)
Residential area ^1^	
Outside Fukushima	178 (67.4)
Fukushima	86 (32.6)
Living	
Alone	47 (17.8)
With a partner only (no children)	76 (28.8)
With children (or children and family)	87 (33.0)
Other	54 (20.5)
**Radiation-related items**	
Living near any NPP (nuclear power plant)^1^	
No	169 (64.0)
Yes	95 (36.0)
Concern related to living near any NPP	
No	45 (17.0)
Yes	219 (83.0)

^1^ Items to set up a target number of research participants, ^2^ The respondents who answered “Full time or Part time” to employment status (n = 167).

**Table 3 ijerph-18-12007-t003:** eHealth literacy levels of research participants.

eHealth Literacy	Mean (SD)	Min	Max	Median (25th–75th Percentile)
*Total score*	*23.8 (5.6)*	*8*	*40*	*24.0 (20.0–28.0)*
1. I know what health resources are available on the Internet.	2.96 (0.95)	1	5	3.0 (2.0–4.0)
2. I know where to find helpful health resources on the Internet.	2.92 (0.92)	1	5	3.0 (2.0–4.0)
3. I know how to find helpful health resources on the Internet.	3.13 (0.91)	1	5	3.0 (3.0–4.0)
4. I know how to use the Internet to answer my health questions.	3.20 (0.89)	1	5	3.0 (3.0–4.0)
5. I know how to use the health information I find on the Internet to help me.	3.08 (0.85)	1	5	3.0 (2.3–4.0)
6. I have the skills I need to evaluate the health resources I find on the Internet.	2.75 (0.95)	1	5	3.0 (2.0–3.0)
7. I can tell high quality from low quality health resources on the Internet.	2.82 (0.89)	1	5	3.0 (2.0–3.0)
8. I feel confident in using information from the Internet to make health decisions.	2.97 (0.89)	1	5	3.0 (2.0–4.0)

The internal consistency of the eight items was adequately high (Cronbach’s α = 0.91).

**Table 4 ijerph-18-12007-t004:** Factors associated with “knowledge, attitude, and practice” regarding a digital application for radiation protection.

	Radiation Assessment Digital Tool [N (%)]								
	Knowledge			Attitude			Practice		
	No [183(69.3)]	Yes [81 (30.7)]	*p* Value	No [73 (27.7)]	Yes [191 (72.3)]	*p* Value	No [233 (88.3)]	Yes [31 (11.7)]	*p* Value
**Individual attributes ^1^**									
Sex									
Female	102 (76.7)	31 (23.3)	0.01	37 (27.8)	96 (72.2)	0.95	125 (94.0)	8 (6.0)	0.004
Male	81 (61.8)	50 (38.2)		36 (27.5)	95 (72.5)		108 (82.4)	23 (17.6)	
Age									
20–59	134 (75.3)	44 (24.7)	0.003	48 (27.0)	130 (73.0)	0.72	162 (91.0)	16 (9.0)	0.046
60 or older	49 (57.0)	37 (43.0)		25 (29.1)	61 (70.9)		71 (82.6)	15 (17.4)	
Education									
High school or lower	66 (66.0)	34 (34.0)	0.47	29 (29.0)	71 (71.0)	0.75	88 (88.0)	12 (12.0)	0.97
Junior college, technical school	48 (75.0)	16 (25.0)		19 (29.7)	45 (70.3)		57 (89.1)	7 (10.9)	
University, graduate school	69 (69.0)	31 (31.0)		25 (25.0)	75 (75.0)		88 (88.0)	12 (12.0)	
Employment status									
Employed	125 (72.7)	47 (27.3)	0.11	45 (26.2)	127 (73.8)	0.46	152 (88.4)	20 (11.6)	0.94
Unemployed	58 (63.0)	34 (37.0)		28 (30.4)	64 (69.6)		81 (88.0)	11 (12.0)	
Job type *									
Engineering, technical expert, managerial work	33 (68.8)	15 (31.3)	0.61	11 (22.9)	37 (77.1)	0.63	42 (87.5)	6 (12.5)	0.93
Administration	30 (71.4)	12 (28.6)		10 (23.8)	32 (76.2)		37 (88.1)	5 (11.9)	
Other	59 (76.6)	18 (23.4)		23 (29.9)	54 (70.1)		69 (89.6)	8 (10.4)	
Residential area									
Outside Fukushima	132 (74.2)	46 (25.8)	0.01	49 (27.5)	129 (72.5)	0.95	168 (94.4)	10 (5.6)	<0.001
Fukushima	51 (59.3)	35 (40.7)		24 (27.9)	62 (72.1)		65 (75.6)	21 (24.4)	
Living									
Alone	37 (78.7)	10 (21.3)	0.03	18 (38.3)	29 (61.7)	0.34	42 (89.4)	5 (10.6)	0.63
With a partner only (no children)	43 (56.6)	33 (43.4)		19 (25.0)	57 (75.0)		64 (84.2)	12 (15.8)	
With children (or children and family)	65 (74.7)	22 (25.3)		23 (26.4)	64 (73.6)		78 (89.7)	9 (10.3)	
Other	38 (70.4)	16 (29.6)		13 (24.1)	41 (75.9)		49 (90.7)	5 (9.3)	
**Radiation-related items ^1^**									
Living near any NPP ^3^									
No	119 (70.4)	50 (29.6)	0.61	53 (31.4)	116 (68.6)	0.07	149 (88.2)	20 (11.8)	0.95
Yes	64 (67.4)	31 (32.6)		20 (21.1)	75 (78.9)		84 (88.4)	11 (11.6)	
Concern related to living near any NPP									
No	34 (75.6)	11 (24.4)	0.32	19 (42.2)	26 (57.8)	0.02	42 (93.3)	3 (6.7)	0.25
Yes	149 (68.0)	70 (32.0)		54 (24.7)	165 (75.3)		191 (87.2)	28 (12.8)	
**eHealth Literacy (eHL) ^2^**									
Mean (SD)	23.1 (5.5)	25.6 (5.5)	0.001	22.4 (6.1)	24.4 (5.4)	0.01	23.6 (5.7)	25.8 (5.0)	0.04

^1^ Chi-square test, Fisher exact test; ^2^ Independent t-test; ^3^ Nuclear power plant; * The respondents who answered “Full time or Part time” to employment status (knowledge no: n = 122, yes: n = 45; attitude no: n = 44, yes: n = 123; practice no: n = 148, yes: n = 19).

**Table 5 ijerph-18-12007-t005:** Factors associated with “knowledge, attitude, and practice” regarding a digital application for radiation protection: multivariable analysis.

	Radiation Assessment Digital Tool								
	Knowledge			Attitude			Practice		
	aOR ^1^	95%CI ^2^	*p* Value ^3^	aOR ^1^	95%CI ^2^	*p* Value ^3^	aOR ^1^	95%CI ^2^	*p* Value ^3^
**eHealth Literacy**									
Total score	1.10	1.04–1.16	0.001	1.06	1.01–1.12	0.02	1.10	1.01–1.19	0.02
**Individual attributes**									
Sex									
Female	1						1		
Male	1.70	0.94–3.09	0.08				3.07	1.23–7.69	0.02
Age									
20–59	1						1		
60 or older	1.71	0.89–3.29	0.11				1.84	0.78–4.32	0.16
Residential area									
Outside Fukushima	1						1		
Fukushima	2.17	1.21–3.91	0.01				6.73	2.85–15.94	<0.001
Living									
Alone	1								
With a partner only (no children)	2.30	0.94–5.63	0.07						
With children (or children and family)	1.31	0.53–3.21	0.56						
Other	1.75	0.67–4.57	0.26						
**Radiation-related items**									
Concern related to living near any NPP ^4^									
No				1					
Yes				2.18	1.11–4.29	0.02			

^1^ Adjusted odds ratio; ^2^ 95% confidence interval; ^3^ Binomial logistic regression analysis, Interest in the radiation digital tool (knowledge, attitude and practice) as the response variable. The response of “No” as reference (=0), “Yes” as 1; ^4^ Nuclear power plant.

**Table 6 ijerph-18-12007-t006:** Factors associated with “knowledge, attitude, and practice” regarding a digital application for health promotion.

	Health Promotion Digital Tool [N (%)]								
	Knowledge			Attitude			Practice		
	No [214 (81.1)]	Yes [50 (18.9)]	*p* Value	No [91 (34.5)]	Yes [173 (65.5)]	*p* Value	No [246 (93.2)]	Yes [18 (6.8)]	*p* Value
**Individual attributes ^1^**									
Sex									
Female	114 (85.7)	19 (14.3)	0.05	46 (34.6)	87 (65.4)	0.97	125 (94.0)	8 (6.0)	0.60
Male	100 (76.3)	31 (23.7)		45 (34.4)	86 (65.6)		121 (92.4)	10 (7.6)	
Age									
20–59	146 (82.0)	32 (18.0)	0.57	66 (37.1)	112 (62.9)	0.20	164 (92.1)	14 (7.9)	0.33
60 or older	68 (79.1)	18 (20.9)		25 (29.1)	61 (70.9)		82 (95.3)	4 (4.7)	
Education									
High school or lower	84 (84.0)	16 (16.0)	0.02	38 (38.0)	62 (62.0)	0.47	94 (94.0)	6 (6.0)	0.84
Junior college, technical school	57 (89.1)	7 (10.9)		23 (35.9)	41 (64.1)		60 (93.8)	4 (6.3)	
University, graduate school	73 (73.0)	27 (27.0)		30 (30.0)	70 (70.0)		92 (92.0)	8 (8.0)	
Employment status									
Employed	140 (81.4)	32 (18.6)	0.85	62 (36.0)	110 (64.0)	0.46	160 (93.0)	12 (7.0)	0.89
Unemployed	74 (80.4)	18 (19.6)		29 (31.5)	63 (68.5)		86 (93.5)	6 (6.5)	
Job type *									
Engineering, technical expert, managerial work	39 (81.3)	9 (18.8)	0.04	21 (43.8)	27 (56.3)	0.12	46 (95.8)	2 (4.2)	0.15
Administration	29 (69.0)	13 (31.0)		10 (23.8)	32 (76.2)		36 (85.7)	6 (14.3)	
Other	68 (88.3)	9 (11.7)		30 (39.0)	47 (61.0)		73 (94.8)	4 (5.2)	
Residential area									
Outside Fukushima	144 (80.9)	34 (19.1)	0.92	54 (30.3)	124 (69.7)	0.04	165 (92.7)	13 (7.3)	0.65
Fukushima	70 (81.4)	16 (18.6)		37 (43.0)	49 (57.0)		81 (94.2)	5 (5.8)	
Living									
Alone	37 (78.7)	10 (21.3)	0.87	19 (40.4)	28 (59.6)	0.48	43 (91.5)	4 (8.5)	0.74
With a partner only (no children)	61 (80.3)	15 (19.7)		29 (38.2)	47 (61.8)		70 (92.1)	6 (7.9)	
With children (or children and family)	73 (83.9)	14 (16.1)		25 (28.7)	62 (71.3)		83 (95.4)	4 (4.6)	
Other	43 (79.6)	11 (20.4)		18 (33.3)	36 (66.7)		50 (92.6)	4 (7.4)	
**Radiation-related items ^1^**									
Living near any NPP ^3^									
No	139 (82.2)	30 (17.8)	0.51	67 (39.6)	102 (60.4)	0.02	162 (95.9)	7 (4.1)	0.02
Yes	75 (78.9)	20 (21.1)		24 (25.3)	71 (74.7)		84 (88.4)	11 (11.6)	
Concern related to living near any NPP (Nuclear power plant)									
No	39 (86.7)	6 (13.3)	0.29	25 (55.6)	20 (44.4)	0.001	43 (95.6)	2 (4.4)	0.75
Yes	175 (79.9)	44 (20.1)		66 (30.1)	153 (69.9)		203 (92.7)	16 (7.3)	
**eHealth Literacy (eHL) ^2^**									
Mean (SD)	23.2 (5.4)	26.5 (5.9)	<0.001	22.6 (6.2)	24.5 (5.2)	0.007	23.6 (5.5)	27.5 (6.7)	0.004

^1^ Chi-square test, Fisher exact test; ^2^ Independent *t*-test; ^3^ Nuclear power plant; * The respondents who answered “Full time or Part time” to employment status (knowledge no: n = 136, yes: n = 31; attitude no: n = 61, yes: n = 106; practice no: n = 155, yes: n = 12).

**Table 7 ijerph-18-12007-t007:** Factors associated with “knowledge, attitude, and practice” regarding a digital application for health promotion: multivariable analysis.

	Health Promotion Digital Tool								
	Knowledge			Attitude			Practice		
	aOR ^1^	95%CI ^2^	*p* Value ^3^	aOR ^1^	95%CI ^2^	*p* Value ^3^	aOR ^1^	95%CI ^2^	*p* Value ^3^
**eHealth Literacy**									
Total score	1.13	1.06–1.20	<0.001	1.06	1.01–1.11	0.01	1.16	1.05–1.29	0.005
**Individual attributes**									
Education									
High school or lower	1								
Junior college, technical school	0.59	0.22–1.55	0.28						
University, graduate school	2.03	0.99–4.16	0.05						
Residential area									
Outside Fukushima				1					
Fukushima				0.65	0.35–1.21	0.17			
**Radiation-related items**									
Living near any NPP ^4^									
No				1			1		
Yes				1.68	0.88–3.19	0.11	3.18	1.16–8.71	0.02
Concern related to living near any NPP ^4^									
No				1					
Yes				3.17	1.59–6.31	0.001			

^1^ Adjusted odds ratio; ^2^ 95% confidence interval; ^3^ Binomial logistic regression analysis, Interest in the radiation digital tool (knowledge, attitude and practice) as the response variable. The response of “No” as reference (=0), “Yes” as 1; ^4^ Nuclear power plant.

## Data Availability

Relevant questionnaire data have been presented herein. Other data are securely protected, but may be made available to qualified researchers upon reasonable request and in accord with local policy, national law, and the World Medical Association Declaration of Helsinki.

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
