# Peer review of "User-Guided Design of a Digital Tool for Health Promotion and Radiation Protection: Results from an Internet Needs Survey"

_ijerph, 2021, doi:10.3390/ijerph182212007_

Round 1

Reviewer 1 Report

In general terms, the theme of the article seems interesting to me, since the value of the importance of media literacy and the use of digital tools integrated in social life and, in particular, in health matters, for the prevention and promotion of health.

Another relevant aspect of the article is that it proposes an extrapolated research model.

However, the parts into which the text is divided seem too brief to me, especially Introduction and References sections.

What's more:

Title. I find it attractive, although I would change the order of radiation protection and health promotion: "User-guided design of a digital tool for health promotion and radiation protection (...)", in the case of that health promotion is related to radiation, which is not clear to me. I think the title is too pretentious and creates expectations that are not fully resolved in the text.

Introduction:

Page 1. “With an increase of global disasters, digital health services, including telemedicine, could improve post-disaster mental health care for under-served populations”: Is there a study on mental health that corroborates this ruling?

Methodology:

Page 2. “Disaster eHealth approaches should thus be incorporated into disaster planning and preparedness so as to be ready for use during and after a disaster”: It is implied that there are no previous experiences on health crisis situations. Perhaps, it should be noted that they “should be incorporated more frequently”.

"Study purpose". In this section the objectives of the study are indicated, without being clearly defined and are mixed with what should be included in the methodology section.

Page 3. “Study participants”. More information would be needed on the selection of the sample: Why was this one selected and not another? Their number could be increased so that the investigation has greater weight.

The states of "concern" and "anxiety" are identified, when they do not have to be similar emotions. In any case, if it is identified, it should be supported by a previous study, for example.

"EHL level was associated with sex, age, employment status, marital status, and income": Needed to explain to what extent.

Results:

Page 4. In table 1, it would be interesting to know separately the anxiety data only for those who live near any nuclear power plant.

Page 5. In table 2, “營”: I don’t identify this symbol.

The content of different sections is mixed: for example, "the result showed that a higher eHL was significantly associated with positive KAP" and "Positive KAP is significantly correlated with higher eHL scores, in contrast to Negative KAP. " are conclusions.

Discussion:

Page 8. Again, content from different sections is mixed. For example, conclusions are introduced that could be drawn for later: “For example, at the family level, we can encourage family members to relate to (...) ”.

Page 9. Furthermore, terms in this section of a more subjective rather than objective nature should not be used: for example, “interesting”.

References:

Page 10. Although most of the references are recent, some, such as number 6 (“Norman, CD; Skinner, HA eHealth Literacy: Essential Skills for Consumer Health in a Networked World. J Med Internet Res 2006, 8, e9. ”) are not (2006), when the subject matter has changed a lot over time.

Reviewer 2 Report

Thank you for the opportunity to review this article. This article is very interesting because it uses a very actual tool, an Internet survey, for making a user-guided design of digital tool for radiation protection. There are some interesting aspects in this research, and I really liked it, albeit there are some aspects that perhaps could be revised:

In Methods, the authors describe that «This online survey was conducted from 31 January to 4 February 2020 with participants recruited from monitors registered with an Internet survey company, INTAGE Inc». This is an essential aspect in this research. Selection bias is unavoidable in web surveys, and the best way to try to avoid it is using tools like CHERRIES methodology and clearly describing the sample. In this case, the authors get the sample from an Internet survey company. Therefore it is easy to understand that this selection bias could be considerable. As it is also unavoidable, I think that the authors perhaps could better describe the population where they obtained their sample, and clearly describe the main features of their sample. Their results and conclusions depend on the composition of this sample, so it is essential that the readers could understand the main features of the sample. Another important aspect, related to this, is describing the use of incentives, if they were used or not. All these aspects, if better or clearly described, could help to better understand the sample and, therefore, the results and the conclusions. I think this is the most important part to revise, as the results and conclusions directly depend on the sample composition.

In Methods, the authors write that « In total, 339 responded, and we extracted the data of 264 lay persons for analyses in the present study». Here, the authors could describe if they performed any kind of sample size estimation, and how. They could also described why they extracted data of only 264 persons.

The authors also write that «In addition to the SHAMISEN-SINGS survey items, we included an eHL measure» and that «we used a Japanese version of the eHealth Literacy Scale». They could described if these questionnaires were previously validated, or if they performed transcultural adaptation (in case it was required). For example, the authors could describe if questions like «Are you personally concerned about potential dangers and risks related to living close to a nuclear power plant?» were previously validated in other questionnaires.

The authors write that «Cronbach’s coefficient alpha was used to measure the internal consistency of the 8 items regarding eHL level». It would be interesting to know if they performed any kind of Confirmatory Factorial Analysis (CFA), and why.

As a consequence of these aspects, the limitations section could better describe some aspects, like better describing the potential selection bias, that does not invalidate the research, but it is unavoidable and perhaps it could be better and clearer described.

In the Conclusions (or in the Discussion), I would invite the authors (this is just a proposal), if they could add any idea, or opinion, on how other users could be attracted to these kind of apps. This could help future researchers, in this field.

Despite these aspects that I think that perhaps could be better described, I think that this is a very interesting research that really helps other researchers, in this specific topic. I really liked it.

Reviewer 3 Report

Dear Authors,

The analyzed article is an interesting and original presentation of the topic.
Substantively important topic deserves to be published in a journal.
I would suggest to slightly refine the initial part of the chapter Introduction in a way that clearly specifies the purpose / research intention of the paper.

The subject matter presented and the coherent presentation of results outweighs the evaluation toward acceptance of the paper.
